# Design and Application of Near Infrared LED and Solenoid Magnetic Field Instrument to Inactivate Pathogenic Bacteria

**DOI:** 10.3390/mi14040848

**Published:** 2023-04-14

**Authors:** Deny Arifianto, Suryani Dyah Astuti, Perwira Annissa Dyah Permatasari, Ilmi Arifah, Ahmad Khalil Yaqubi, Riries Rulaningtyas, Ardiansyah Syahrom

**Affiliations:** 1Faculty of Science and Technology, Airlangga University, Surabaya 60115, Indonesia; 2Department of Physics, Faculty of Science and Technology, Airlangga University, Surabaya 60115, Indonesia; 3Department of Mathematics, Faculty of Science and Technology, Airlangga University, Surabaya 60115, Indonesia; 4Medical Devices and Technology Centre, Universiti Teknologi Malaysia, Bahru, 81310, Malaysia

**Keywords:** *Escherichia coli*, photoinactivation, Infrared LED, solenoid magnetic field, *Staphylococcus aureus*

## Abstract

Purpose: This study aims to evaluate the efficiency of infrared LEDs with a magnetic solenoid field in lowering the quantity of gram-positive *Staphylococcus aureus* and gram-negative *Escherichia coli* bacteria, as well as the best exposure period and energy dose for inactivating these bacteria. Method: Research has been performed on a photodynamic therapy technique called photodynamic inactivation (PDI), which combines infrared LED light with a wavelength range of 951–952 nm and a solenoid magnetic field with a strength of 0–6 mT. The two, taken together, can potentially harm the target structure biologically. Infrared LED light and an AC-generated solenoid magnetic field are both applied to bacteria to measure the reduction in viability. Three different treatments infrared LED, solenoid magnetic field, and an amalgam of infrared LED and solenoid magnetic field, were used in this study. A factorial statistical ANOVA analysis was utilized in this investigation. Results: The maximum bacterial production was produced by irradiating a surface for 60 min at a dosage of 0.593 J/cm^2^, according to the data. The combined use of infrared LEDs and a magnetic field solenoid resulted in the highest percentage of fatalities for *Staphylococcus aureus*, which was 94.43 s. The highest percentage of inactivation for *Escherichia coli* occurred in the combination treatment of infrared LEDs and a magnetic field solenoid, namely, 72.47 ± 5.06%. In contrast, *S. aureus* occurred in the combined treatment of infrared LEDs and a magnetic field solenoid, 94.43 ± 6.63 percent. Conclusion: *Staphylococcus aureus* and *Escherichia coli* germs are inactivated using infrared illumination and the best solenoid magnetic fields. This is evidenced by the rise in the proportion of bacteria that died in treatment group III, which used a magnetic solenoid field and infrared LEDs to deliver a dosage of 0.593 J/cm^2^ over 60 min. According to the research findings, the magnetic field of the solenoid and the infrared LED field significantly impact the gram-positive bacteria *S. aureus* and the gram-negative bacteria *E. coli*.

## 1. Introduction

Microorganisms that cause contamination greatly affect things that should be kept clean, such as surgical instruments and medical equipment. *Staphylococcus aureus* and *Escherichia coli* are two contaminating microorganisms frequently discovered on medical equipment [1]. Sterilization is a technique for handling tools or materials that eliminatesall microbiological life, including bacteria. It can be achieved via chemical or physical techniques. Medical equipment can be sterilized in several ways, including through chemical procedures such as disinfection techniques and physical methods such as filtering, heating, and radiation [2]. When heating techniques are used in the physical method, they can make it easier for metals to dissolve in the carrier material. This makes it harder for bacteria to live in the carrier material. Numerous sterilizing devices are on the market today, each with a unique working mechanism and a set of restrictions. One alternative way to kill germs that have been contaminated that works well is the photoinactivation method.

Recently, photoinactivation has become an important new medical therapy for treating bacterial infections that won’t go away. Photoinactivation is a part of photodynamic therapy. It makes free radicals, specifically reactive oxygen species, which can kill microbial cells [3,4,5]. The existence of a light source, photosensitizers, and free radicals are the three elements that determine whether the photodynamic process for cell inactivation is successful [6]. Numerous bacterial species naturally synthesize endogenous porphyrins, which are photosensitizing, light-absorbing molecules. Photosensitizers can absorb certain wavelengths of light, and they can also generate reactive oxygen species (ROS). The photosensitization mechanism initiates the photoinactivation process. The photophysical process’s interaction between light and the photosensitizer is a physical phenomenon in the photosensitization mechanism [3]. Photophysical activities take place in the photosensitizer molecule’s electron level area, where each molecule contains an electron with a spin pair at the singlet state level. The first thing that happens when a molecule is exposed to light is that it absorbs light [7]. The majority of absorbing molecules will become stimulated. Because it is reactive and can combine with nearby molecules such as lipids or oxygen to form reactive oxygen, the triplet state level is good for an excited state [3]. This reactive oxygen, also called radical oxygen species (ROS), can stop microbial cells from working when exposed to light [8].

The parietal structure of bacteria may be harmed due to exposure to pulsed light. The light-emitting diode is one device that employs pulsed light [9]. Diodes that can emit light with a wavelength between 700 nm and 106 nm are known as infrared LEDs. It is a property of infrared LEDs that they are quickly absorbed by organic substances [10]. The infrared LEDs can kill germs at a wavelength of 950 nm. The physical quantities of the light source that play an important role in the photoinactivation process are the energy intensity and dose. These quantities can be expressed in Equations (1) and (2).
(1)$$I=\frac{P}{A}$$
(2)$$D=I.t=\frac{P\;t}{A}$$
where *A* is the area of the beam (in cm^2^), *D* is the energy density of the LED irradiation (in J/cm^2^), I is the intensity (in W/cm^2^), P is the power (in W), t is the irradiation period (in s), and P is the power.

During irradiation, the porphyrin molecule takes the photon, which then moves to a higher energy state. Molecules can excite from the electronic singlet S0’s ground state to its excited state due to their absorption of photons. The ground state of a molecular excitation leading to a higher energy state will often be reached through radiative and non-radiative transitions. When the energy level of the singlet excited state coincides with that of the triplet excited state, putting the electron in the triplet state, the spin of an excited singlet electron Sn can be reversed (intersystem crossing). Photochemical reactions take place in this triplet state, resulting in the production of numerous reactive oxygen species. It’s possible that variations in the magnetic dipole moment will cause the spin-orbit coupling process, which will lead to the intersystem crossing process. The interplay of orbital and spin momentum, which results in total angular momentum, changes the dipole moment. The sharing of energy levels known as the Zeeman effect, one of which overlaps with the singlet excitation, will occur if there is an external magnetic field because this total angular momentum will align in the direction of the magnetic field [11].

When light and a magnetic field are coupled in photodynamic therapy, pathogenic bacteria will be photoactivated more quickly [12]. Magnetic fields can increase energy by producing a strong magnetic field during photosensitization [3]. The magnetic field is a physical quantity that expresses the state of matter in a substance or location. The area still subject to the magnetic force is the magnetic field. When the medium substrate is exposed by the light source and magnetic field, the number of bacterial cells can change depending on spectrum, energy density of light, and magnetic field [13].

When the same parameters for the exposure time and the addition of the oxygen ratio were used in the absence of the magnetic field, a dark circle in the center showed no growth regions, as shown in Figure 4 for an exposure time from 2 min to 10 min. Around these regions, a dense mat of bacterial growth covers the agar and masks the dark background. We noticed that the effect of plasma discharge on *E. coli* increased at a small exposure time, approximately half the value at the time needed for the inactivation process in the absence of the magnetic field. Furthermore, the area of the inactivated region was much greater when oxygen was added to argon using Ar/3% O_2_.

The movement of electric charges creates a magnetic field. In microscopically magnetic materials on the atomic scale, tiny currents happen due to electron rotation or circulation around the atomic nucleus [14]. A solenoid magnetic field is one kind of magnetic field that results from current flow. The conductor wire is wrapped into a cylinder as part of the procedure that creates the solenoid. Along the coil’s axis, the lines of magnetic lines of force are spread uniformly and in parallel, and their distribution inside the coil is similar. Figure 1 displays the DC-current-driven solenoid magnetic field lines.

The Poynting vector is the term used to characterize electromagnetic energy density. The propagation of electromagnetic energy in a path normal to the directions of the electric and magnetic fields is described as this physical quantity [16]. The Poynting vector is denoted mathematically as:(3)$$\vec{S}=\frac{1}{\mu_{0}}\;\vec{E_{0}}\;\times \;\vec{B_{0}}$$
where $\;\vec{S}$ is the Poynting vector, which stands for the physically quantifiable property of the electromagnetic field’s intensity. According to Equation (3), the electromagnetic energy density moves in space-time in a direction corresponding to wave propagation. It is proportional to the square of the amplitude.

The propagation of electromagnetic waves in a conducting substance suffers amplitude attenuation as a function of the penetration distance to the material, in contrast to propagation in a vacuum or air. This is expressed mathematically as Equations (4) and (5) below.
(4)$$\vec{E}=\vec{E_{0}}e^{-KZ}\sin\left(kz\;\pm \;\omega t\right)$$
(5)$$\vec{B}=\vec{B_{0}}e^{-KZ}\sin\left(kz\;\pm \;\omega t\right)$$

$\vec{E_{0}}$ and $\vec{B_{0}}$ are the electric and magnetic field amplitudes, respectively, while $k=\frac{2\pi }{\lambda }\;$ is the wave propagation constant, where λ is the wavelength, ω=2πv is the wave oscillation speed, and v is the wave frequency. The equation c=vλ, which demonstrates a clear exponential decline in amplitude in the direction of wave propagation, relates wavelength and wave frequency. Magnetic field-induced structural disorder in bacterial cells is correlated with morphological and mechanical alterations of the cell wall [17]. Bacteria will have a rough surface and demolish the cell wall, causing harm to the cell membrane. As a result, the cytosol and bacterial DNA were lost. Meanwhile, alternating magnetic field therapy can decrease cell viability [18].

The method that has been used to inactivate bacteria is the use of antibiotics. However, the use of antibiotics causes bacteria to become resistant. For example, Methicillin Resistant *Staphylococcus aureus* (MRSA) is a *Staphylococcus aureus* bacterium resistant to the antibiotic methicillin [19]. MRSA can form self-protection against antibiotics by forming a biofilm layer, which is a community of bacterial cells that are structured and stick together to form colonies that are capable of producing a hydrated polymer matrix of exopolymer substances, polysaccharides, nucleic acids, and proteins on biotic or abiotic surfaces [20]. Bacteria that are already resistant to antibiotics will be difficult to treat. Photodynamic Inactivation (PDI) is a method used to inactivate microbes by irradiating photons of light. There are three main factors at play in the success of PDI––a light source, a photosensitizer (PS), and free radical products that are reactive to biological systems such as cells. Various light sources have been used in PDI, including the light-emitting diode (LEDs) [21,22].

By dividing the emitted LED power by the beam area (W/cm^2^), one can calculate the amount of light that is present. Radiant flux is generally measured using an integrating sphere, but simple measurements can be made using a reflector (mirror) instrument. The light emitted from an LED is not a single wavelength but is polychromatic, so the amount of light from each wavelength must be measured and integrated to get the right emission flux value. In most cases, however, the photodiode current is measured and converted to a light level based on the spectral response of the photodiode to the peak emission wavelength [23]. This method is similar to using LEDs, so measuring the flux with this method presents no significant problems.

It has been demonstrated that exposing bacteria to LED light irradiation and a magnetic field may help lower the proportion of bacterial colonies [24]. It has been demonstrated that when the two are combined, the bacteria are already reduced by 70–80%. The best setup for photoinactivation of water-polluting bacteria is a green LED with a wavelength of 541 nm at an irradiation energy dose of 15 J/cm^2^ and a low-frequency magnetic field strength of 1.8 mT (*Escherichia coli*) [3].

A significant difference between gram-positive and negative bacteria lies in the structure of their biological composition, so the effect obtained from the combination therapy between LED light and a magnetic field on the inactivation of gram-positive and negative bacteria is also different [25]. The distinguishing characteristic of gram-positive bacteria is the composition of their cell wall. Several layers of peptidoglycan join together to form a thick, rigid structure. About 40 peptidoglycan layers, also called murein or mucopeptide layers, make up 50% of the cell wall material. In Gram-negative bacteria, there are only one or two layers, constituting 5–10% of the cell wall material. In addition, the cell wall of gram-positive bacteria contains teichoic and glucuronic acids, which are mainly composed of alcohols (such as ribitol and alcohol) and phosphates. There are two types of teichoic acid: lipoteichoic acid and wall teichoic acid. Both types of teichoic acids are negatively charged because they contain a phosphate group in their molecular structure.

## 2. Materials and Methods

### 2.1. Infrared LED

In this study, the characterization process of infrared LEDs was carried out to establish the LEDs’ specifications and acquire the ideal power for killing Escherichia coli and *Staphylococcus aureus* germs. The Laboratory of Biophysics and Medical Physics, Physics Study Program, Faculty of Science and Technology, Airlangga University, used a power meter to characterize the power of infrared LEDs. The tool increases the infrared LED’s output power and wavelength values in this case. To read the power value regarding the location of the *Staphylococcus aureus* and *Escherichia coli* bacteria samples, the light beam from the source (LED) is fired at a distance of 13 cm from the power meter detector sensor, with the results obtained being displayed by the display connected to the detector. A time range of 0 to 70 min was used for the temperature characterization. The intensity of the LED is then extensively measured, specifically using the scanning method. The location of each scanning procedure is determined by the overall position of the scale board created earlier. This scale board is used to examine the distribution of uniform power values that can be utilized as a sample site and for therapy, as well as the distribution of measured power values. By summing up all the measured power in the sample area and comparing it to the area of the sample area that forms a circle, the power value acquired is used to calculate the intensity value.

### 2.2. Solenoid Magnetic Field

The solenoid is the magnetic field’s chosen source in this study. The solenoid has 382 turns and is 12.00 ± 0.05 cm long. The principal location for the sample to be placed while describing the magnetic field is determined by measuring the magnitude of the magnetic field strength, which is used to determine the uniformity of the field. Figure 2 illustrates the magnetic field’s characteristics. The stability of the magnetic field strength value is also calculated using the magnetic field’s characteristics. Using a digital tesla meter with a precision of 0.01 mT, the magnitude of the magnetic field value was measured.

### 2.3. Mechanical Design

The main frame is built using aluminum profiles covered with an acrylic sheet. The movement of the z-axis uses a screw mechanism connected to a stepper motor and two linear bearings as the guide shaft. This method allows the distance of the LED array from the sample area to be varied between 1 cm to 15 cm. The system design is shown in Figure 3.

### 2.4. Hardware Design

This system consists of a microcontroller circuit, a temperature sensor (DS18B22), a relay circuit to activate the solenoid array, an LED driver circuit to activate the LED array, and an LCD touchscreen display. The block diagram of the microcontroller-based CNC laser diode is shown in Figure 4.

### 2.5. Software Design

Software design includes DS18B22 temperature sensor initialization, driver stepper motor initialization, exposure time, LED array height, combination LED and solenoid mode, and graphical user interface (GUI) design. The design software is shown in Figure 5.

### 2.6. Temperature Stability

A digital thermometer was used to characterize the temperature at the Biophysics and Medical Physics Laboratory, Physics Study Program, Faculty of Science and Technology, Airlangga University (thermogenic). *Staphylococcus aureus* and *Escherichia coli* bacteria samples are shot by the thermogenic at a distance of 1 cm, and the findings are then shown on display. Temperature characterization is conducted to determine if the temperature will be steady and unaffected by the environment’s temperature during the therapeutic procedure. The time range used for the characterization was 0 to 70 min.

### 2.7. Bacterial Culture

The bacterial strains *Staphylococcus aureus* ATCC 25923 and Escherichia coli ATCC 25922 were inoculated on Tryptone Soy Agar (Oxoid, Basingstoke, UK) and taken on Tryptone Soy Broth (Oxoid, Basingstoke, UK). The culture of bacteria was incubated at 37 °C until bacterial colonies reached ~108 CFU/mL or 1.0 McFarland Standard.

### 2.8. Treatment of the Sample

*S. aureus* and *E. coli* bacterial culture samples were divided into four groups: control group K without treatment, group T1 treated with infrared LED exposure, group T2 with magnetic field treatment, and group T3 with infrared LED 951 nm treatment and magnetic field treatment. Each group was treated with various variations, with a treatment time of 30, 40, 50, and 60 min. There were 16 experimental units, with a total of 5 replications for each treatment group. After treatment, the samples were given agar media and incubated for 24 h. Furthermore, the number of bacterial colonies that grew with each treatment was counted.

### 2.9. Data Analysis

Data analysis used a factorial two-way ANOVA test to determine the effect of treatment on each group. The percentage of bacterial inactivation was then calculated based on the following Equation (6).
(6)$$p=\left|\frac{\sum koloni\;kontrol-\sum koloni\;perlakuan}{\sum koloni\;kontrol}\right|\;\times \;100\%$$

## 3. Results

### 3.1. Infrared LED Power Characterization Test Results

#### 3.1.1. Characterization of Power in the Sample Space

The power characterization in this sample room aims to determine the distribution of the output power given by the infrared LED to the position of the sample. In the power characterization stage, a scale board was made using millimeter block paper with a size of 20 cm × 10 cm, which was then divided into 10 squares × 5 squares, as shown in Figure 3, with each box measuring 2 cm × 2 cm. Next, each box scans the infrared LED output power value using a power meter to determine the power value. The power chosen as the sample position is uniform or has no significant difference. The selected sample positions include nine boxes (P3,1; P3,2; P3,3; P4,1; P4,2; P4,3; P5,1; P5,2; and P5,3), as shown in Figure 6. Table 1 displays the power distribution results in the sample space chosen in the nine boxes.

A different test is performed to demonstrate that the power values in the chosen sample space do not fluctuate significantly, since it is crucial to demonstrate that the power is distributed uniformly throughout the sample space. This demonstrates that the power is dispersed equally throughout the sample space. As seen in Figure 6, the different test is conducted by separating into two blocks and utilizing the *t*-test. The power distribution in the sample space is uniform and not significantly different when the results have a significance value of *p* = 0.064 > α = 0.05, indicating no difference in the distribution of power in the sample space.

#### 3.1.2. Characterization of Power against Time

The goal of characterizing LED power over time is to establish the stability of LED power from the detector over a given period. Once every three minutes, power measurements were taken until discrete data was gathered. A power meter with units of W in the range of 0–70 min was used to measure power. The distance between the power meter’s detector and the infrared LED was 13 cm when power characterization was carried out. The graph plot of infrared LED power stability versus time is shown in Figure 7.

The procedure of measuring LED power is achieved at each place (P3,1; P3,2; P3,3; P2,1; P2,2; P2,3; P 1,1; P1,2; and P1,3) and successively is (337.29 ± 2.45) W, (325.65 ± 1.78) W, (276.00 ± 1.19) W, (335.16 ± 1.75) W, and (382.27 ± 1.60) According to the findings for each position in the sample room, positions P1,1; P1,2; and P1,3 in Figure 4 have the highest levels of power stability. The graph in Figure 3 can display the three spots’ power stability. The 346–356 W value range is the most stable for LED power. The LED power will be stable during irradiation if the standard deviation is low.

### 3.2. Calculation Results of Infrared LED Energy Dose and Intensity Values

The primary characteristics of the photodynamic treatment, energy intensity, and dose can be calculated based on the findings of power measurements. The power that strikes the sample (cup) is used to calculate the intensity value, which is then compared to the area exposed to the LED. Since the sample is in a circular cup with a diameter of 5 cm, that region is the one that is being used. The LED intensity is obtained to compare the amount of power distribution in the sample space per unit area of the cup. By multiplying the intensity value by the time unit established at the moment of treatment, the results of calculating the intensity value are also used to calculate the value of the energy dose or LED light source energy received by the sample. Table 2 displays the outcomes of estimating energy intensity and dose.

### 3.3. Solenoid Magnetic Field Strength Characterization Test Results

#### 3.3.1. Characterization of Magnetic Field Strength in the Sample Room

The distribution of the solenoid magnetic field values to be employed in the sample room will be determined by the characterization of the solenoid magnetic field strength in the room. An artificial scale board is utilized, just like in power measurements. A Tesla meter scans the magnetic field value in each box on the scale board. For use as the sample position, a magnetic field that is uniform or does not significantly differ is chosen. As illustrated in Figure 7, the chosen sample positions consist of nine boxes (B3,1; B3,2; B3,3; B4,1; B4,2; B4,3; B5,1; B5,2; and B5,3). The outcomes of the power distribution in the sample area selected in the nine boxes are shown in Figure 8 and Table 3.

The magnetic field is spread out evenly in the sample space because it must be shown that the magnetic field chosen for the space is uniform. To do this, a difference test demonstrates that the magnetic field value in the chosen sample space does not have a significant difference. The *t*-test was used to conduct the various tests. The results of the various tests yielded a significant value of *p* = 0.249 > α = 0.05, indicating that the field value at the chosen position has either the same meaning as before or is very similar to it.

#### 3.3.2. Characterization of Magnetic Field Strength against Time

Determine the stability of the magnetic field strength over a specific period by describing the magnetic field. Since the magnetic field being measured is the solenoid, a telemeter with an AC is used to measure the magnetic field’s strength (alternating). The magnetic field strength was recorded every 3 min during the measurement period of 0 to 70 min. The results are erratic because the magnetic field is a solenoid with an AC source. The strong magnetic field is shown to be stable at specific locations, as shown in Figure 9. The stability value of the magnetic field is determined at a value of 4 mT with a measurement error value of 0.03 based on the characteristics of the magnetic field strength.

At each location (B 3,1; B 3,2; B 3,3; B 2,1; B 2,2; B 2,3; B 1,1; B 1,2; and B 1,3), the magnetic field is measured in the following order: (1.45 ± 0.08) mT, (2.21 ± 0.18) mT, (1.38 ± 0.09) mT, (1.33 ± 0.12) mT, (3.45 ± 0.05) mT, (4.19 ± 0.19) mT), (2.21). Figure 9 shows that the magnetic field’s stability level is lower as a result of the findings. This is because the magnetic field alternates with a solenoid magnetic field. The measurement value of the magnetic field’s magnitude is obtained up and down, as seen in Figure 9. According to Figure 9, the matrix positions for B6, B9, and B5, which correspond to the highest scores, are B2,3; B2,2; and B1,3. The LED power will be stable during irradiation if the standard deviation is low.

### 3.4. Temperature Characterization Test Results

The temperature versus time characterization aims to determine if the therapeutic device’s internal temperature is stable. The room’s temperature will continue to have an impact on the device if it is unstable. This measurement was made using a thermometer on the sample area exposed to the light source at predetermined intervals with a time range of 0–70 min. Every minute, the temperature is recorded. The temperature characterization findings are based on the measurements taken, as shown in Figure 10, which is a graph of the temperature characterization results against time.

Figure 10, steady at 29 °C, illustrates how the measurement findings demonstrate that the irradiation temperature is relatively stable. In this case, the temperature in the therapeutic device is not affected by the room temperature because the measured temperature is stable.

### 3.5. Results of Treatment of Staphylococcus aureus

For each treatment, exposure lasted for a varied time for example 30, 40, 50, or 60 min. A new energy dose value is determined for each treatment and variation delivered. There are three treatment groups in the ongoing treatment: solenoid magnetic fields, infrared LED exposure, and a combination. The percentage decrease in *Staphylococcus aureus* colonies vs. the control indicates the effectiveness of the photoinactivation method.

Based on these data, Figure 11 shows the link between the quantity of *Staphylococcus aureus* bacterial colonies and the passage of time. This relationship allows for determining the best moment to cause the largest reduction in bacterial viability. From the graph, it can be inferred that *Staphylococcus aureus* vitality would decrease the longer the treatment process was carried out. As a result, this period serves as a benchmark for a successful treatment procedure. Each time has a fixed value in control (where no treatment is applied). This happened because the control treatment, designed to stop bacterial growth from happening, did not receive the media. As a result, the value of the control obtained did not affect the number of colonies.

Equation (6) can be applied to data on the number of colonies to obtain the proportion of inactivations. The correlation between time and the proportion of *Staphylococcus aureus* bacterium inactivation is shown in Figure 8. According to the calculation, *Staphylococcus aureus* inactivates at a higher percentage the longer the treatment method is conducted. By employing statistical analysis, namely, the two-way factorial ANOVA test, the collected resultant data can be validated to support these findings. If the data are normally distributed, on a minimal interval scale, and have homogeneous variance, the two-way factorial ANOVA test is used to determine the impact of each factor. The findings of the three treatments of the data normality test yielded a significance value of *p* = 0.200 > α = 0.05, indicating that the data are normally distributed. Following a homogeneity test, the data were determined to be homogeneous by the statistical analysis, which produced a significance value of *p* = 0.179 > α = 0.05.

As for the three treatment groups, the two-way ANOVA factorial test results revealed that the significance value achieved was *p* = 0.047 < α = 0.05, indicating a significant difference from each time variation provided. A post hoc test determined which treatment group provided a different meaning. The 50- and 60-min time variations showed considerably different outcomes than the 30- and 40-min ones, which showed no significant difference.

According to statistical test findings, the computation is accurate because treatment group III (which combines an infrared LED with a solenoid magnetic field) has a high percentage of *Staphylococcus aureus* germs that die within 60 min. There is evidence of similarity in the outcomes of the two analyses, as displayed in Table 4. Figure 12 displays the computation’s outcomes in the interim.

### 3.6. Escherichia coli Bacterial Treatment Results

For each treatment, exposure lasted for a varied time, 30, 40, 50, or 60 min, for example. A new energy dose value is determined for each treatment and variation delivered. There are three treatment groups in the ongoing treatment: solenoid magnetic fields, infrared LED exposure, and a combination. The percentage drop in *Escherichia coli* bacteria colonies compared to the control indicates the effectiveness of the photoinactivation process.

Based on these findings, it is possible to see how the time in Figure 13 and the quantity of *Escherichia coli* bacterial colonies relate to one another. According to this correlation, summer is the ideal season to cause the greatest decline in bacterial viability. From the graph, it can be inferred that the *Escherichia coli* bacteria’s vitality would decrease the longer the treatment process was carried out. As a result, this period serves as a benchmark for a successful treatment procedure. With *Staphylococcus aureus*, the situation is the same: in control (without any treatment), the value remains constant. As a result, neither an increase nor a decrease in the number of colonies affects the value the control has acquired.

Equation (6) can calculate the mortality proportion based on information about the number of colonies obtained. The correlation between time and the percentage of Escherichia coli bacterial mortality is shown in Figure 13. According to the calculation, the longer the treatment process, the more *Escherichia coli* germs will inactivate.

By employing statistical analysis, namely, the two-way factorial ANOVA test, the collected resultant data can be validated to support these findings. Suppose the data are normally distributed on a minimal interval scale and have homogeneous variance. In that case, the two-way factorial ANOVA test is used to determine the impact of each factor.

The results of the data normality test from the three treatments obtained a significance value of *p* = 0.200 > α = 0.05, which indicates that the data is normally distributed. A data homogeneity test was then carried out, and from the statistical analysis results, a significance value of *p* = 0.471 > α = 0.05 was obtained, which indicates that the data is homogeneous.

Meanwhile, the two-way ANOVA factorial test results showed that the significance value obtained was *p* = 0.000 < α = 0.05 for the three treatment groups, so it can be concluded that there was a significant difference from each time variation given. A post hoc test determined which treatment group provided a different meaning. The results obtained for the 30-, 40-, 50-, and 60-min variations significantly differed.

The percentage of inactivation of large *Escherichia coli* bacteria obtained in treatment group III (combination of infrared LED with solenoid magnetic field) with a time of 60 min has been proven appropriate by statistical test results. Table 5 contains statistical tests that demonstrate the similarity between the results of the two analyses. Meanwhile, the calculation results are presented in Figure 14.

## 4. Discussion

A technique for photoinactivation bacteria known as photodynamic treatment uses light photons and substances known as photosensitizers as light absorbers. In photodynamic therapy, the interaction of light and chemicals (photosensitizers) results in the photoinactivation of bacteria, characterized by the inhibition of cell metabolic activity due to cytoplasmic membrane damage brought on by reactive oxygen species (ROS) [3]. What makes photodynamic treatment work is the presence of three important components: the light source, the photosensitizer as a substance, and free radicals that kill cells.

Porphyrin chemicals, naturally present in bacteria and acting as endogenous photosensitizers sensitive to light, are found right in the cytoplasm (fluid inside the cell). By using the proper dose of irradiation energy and a wavelength spectrum that matches the porphyrin photosensitizers’ absorption spectrum, it is possible to photoinactivation bacterial cells [9].

LED light exposure damages the bacterial parietal structure [26]. The ability of LEDs to emit or release photon energy justifies their use. The photoinactivation technique uses a wavelength between 951 and 952 nm, within the range of infrared LEDs [27]. Because they offer beneficial properties, such as a light penetration value that is quite deep and easily absorbed by organic materials, infrared LEDs are used [9]. During the irradiation process, the organic parts of the bacterium will quickly absorb the infrared light, which will raise the temperature of the bacteria even more quickly [28].

LEDs that emit infrared light can penetrate farther than visible light. When both light sources are in the same energy state, infrared LEDs’ penetration depth is greater than visible light. The penetration depth will be greater if the wavelength value is higher in the same energy state. The amount of light that passes through tissues varies [29]. These variations depend on the conductivity and permeability values within the cell. Even though the wavelength value of the infrared LED is greater than that of visible light, the depth of penetration produced is greater when considering the intensity, conductivity, and permeability factors. The penetration depth also depends on the intensity value the infrared LED gives to tissue in bacteria.

Due to the horizontal position of the solenoid magnetic field used in this study, the magnetic field strength value does not directly impact the sample through the bottom, but rather, in this instance, due to the use of AC, or alternating current, the magnetic field strength generated has a similar direction, with the direction of the magnetic field alternating to the right and left of the sample. The magnetic field’s strength is inversely proportional to the square of the sample’s distance, which depends on the sample’s location. Because the field does not strike the sample directly from below but rather through the sample’s right and left directions, the value of the magnetic field that impacts the sample in this investigation is less. An AC can more significantly impact the oscillations brought on the magnetic field than by the treatment used in this investigation. Changes in the magnetic field kill bacteria in a significant way.

Because an electric current flowing through it will radiate characteristics resembling those of a magnet, the solenoid connects the current and the magnetic field, which involves atomic currents in the magnet. A magnetic field oscillation happens when the AC (alternating current) current used in this study causes the magnet’s magnetic field to move back and forth.

By transferring energy from the magnetic field to ions in acid-forming bacterial cells, oscillations brought on by magnetic fields can impede the metabolic activity of acid-forming bacteria. In cyclotron resonance, energy is precisely transferred from the magnetic field to the ions in the bacterium. Additionally, energy is transported to ion-related metabolic processes. The ions’ movement and velocity through the cell membrane increase due to the energy transfer to them. It will ultimately cause the cell’s proteins to become damaged. A magnetic field causes damage to proteins that are frequently employed as organic nutrients or cell nutrition, and are important for cellular growth and metabolic functions. These cells’ metabolic functions are inhibited due to damage to their proteins, which disrupts bacterial action [30].

The interaction between the magnetic field and light happens during irradiation. Porphyrin molecules, notably those in the cytoplasmic fluid of bacteria, absorb photons during radiation exposure and are excited from a low energy level to a higher energy level. Twelve molecules can excite from the electronic singlet S0’s ground state to its excited state due to their absorption of photons. The ground state of a molecular excitation leading to a higher energy state will often be reached through radiative and non-radiative transitions.

When the energy level of the singlet excited state coincides with that of the triplet excited state, putting the electron in the triplet state, the spin of an excited singlet electron S n can be reversed (intersystem crossing). Photochemical reactions take place in this triplet state, resulting in the production of numerous reactive oxygen species. Variations in the magnetic dipole moment can cause the spin-orbit coupling process, which in turn can cause the intersystem crossing process. The interplay of orbital and spin momentum, which results in total angular momentum, changes the dipole moment. The sharing of energy levels known as the Zeeman effect, one of which overlaps with singlet excitation, will occur if there is an external magnetic field because the total angular momentum will align in the direction of the magnetic field. A physics equation known as the Poynting vector has been employed to support the relationship between light and the parameter used, namely, the strength of the magnetic field. The Poynting vector is the term used to characterize electromagnetic energy density [31]. The propagation of electromagnetic energy in a path normal to the directions of the electric and magnetic fields is described as this physical quantity. In this study, a solenoid coil that runs AC through its winding to produce oscillations in the magnetic field surrounds it to produce the oscillations. Equation (5) illustrates the mathematical relationship between the magnetic field’s oscillation and wavelength. Bacterial cells exposed to a magnetic field result in structural disarray correlated with modifications to the cell wall’s morphology and mechanics. Bacteria will have a rough surface and demolish the cell wall, causing harm to the cell membrane. As a result, the cytosol and bacterial DNA were lost. Treatment using an alternating magnetic field can reduce cell viability [15].

Gram-positive and gram-negative bacteria were used in this investigation, *Staphylococcus aureus* and *Escherichia coli*. Time variations of 30 min, 40 min, 50 min, and 60 min of treatment were used. When the irradiation time varied, the number of bacterial colonies decreased while the percentage of deaths rose, according to the analysis of the study’s data. The amount of reactive oxygen created increases with time utilized and can cause a large number of bacteria to perish.

Based on these findings, it was determined that an energy dose of 0.593 J/cm^2^ and a stable power from the characterization results caused the highest bacterial inactivation in 60 min. In *Staphylococcus aureus*, the percentage of bacterial inactivation was 93.67% with solenoid magnetic field treatment, 59.88% with infrared LED treatment, and 94.43% with magnetic fields and infrared LED treatment combined. For *Escherichia coli* bacteria, a magnetic solenoid field resulted in a percentage of bacterial mortality of 63.42%, an infrared LED resulted in a percentage of bacterial inactivation of 70.82%, and a combination of a magnetic field and an infrared LED resulted in a percentage of bacterial inactivation of 72.92%.

The findings of the percentage of inactivation for the two bacteria show that *Staphylococcus aureus* produced the highest percentage of inactivation, or 94.43% when treated with an infrared LED and a permanent magnetic field simultaneously for 60 min. Escherichia coli germs, on the other hand, were responsible for 72.92% of all bacterial inactivations. *Escherichia coli* is a gram-negative bacterium, and *Staphylococcus aureus* is a gram-positive bacterium. The cell walls of the two bacteria differ in composition.

The cell wall of gram-positive bacteria is made up of peptidoglycan, teichoic acid, and neuronic acid, and it is thick. Gram-positive bacteria have a single-layered plasma membrane. Most gram-positive bacteria have polysaccharide cell walls, which are more easily harmed by the photoinactivation procedure. Gram-negative bacteria have a double membrane system with a thin peptidoglycan layer between the outer and inner membranes. In contrast, a permeable outer membrane shields their plasma membrane. Layers of phospholipids, lipopolysaccharides, lipoproteins, and proteins comprise the asymmetric outer membrane, which serves as a molecular filter to prevent molecules from the outside from easily entering [11].

After receiving the manual computation results, a statistical analysis was undertaken to determine the best way to distribute the data. The *Staphylococcus aureus* produced significantly better results than the other bacteria, with the largest result being 93.45%, as shown in Table 4. At the same time, the infrared LED has a greater impact on *Escherichia coli’s* mortality rate than the magnetic field, with a resultant figure of 71.31%. Therefore, the magnetic field strength produced by the solenoid coil with AC flowing has a bigger influence than the infrared LED.

Compared to the other two treatment groups, the irradiation outcomes in the treatment group using a combination of magnetic fields and infrared LEDs delivered the highest percentage of bacterial inactivation. This is feasible because reactive oxygen, which infrared LEDs and magnetic fields produce, can damage and lyse bacteria’s biological systems. More reactive oxygen is created, which kills more bacteria.

## 5. Conclusions

An infrared LED was used to kill *Staphylococcus aureus* and *Escherichia coli* germs by putting them in an optimal solenoid magnetic field. The rise in the proportion of bacteria inactivated due to the combination of infrared LEDs and a magnetic solenoid field in treatment group III. Infrared LEDs have the best effects on gram-positive bacteria such as *S. aureus*, whereas gram-negative bacteria such as *E. coli* have the best effects that outperform the solenoid’s magnetic field. In treatment group III, 60 min at a level of 0.593 J/cm^2^ has the best chance of killing *Staphylococcus aureus* and *Escherichia coli* bacteria, based on the percentages of each bacterium being 94.43% and 72.47%, respectively.

## Figures and Tables

**Figure 1 micromachines-14-00848-f001:**
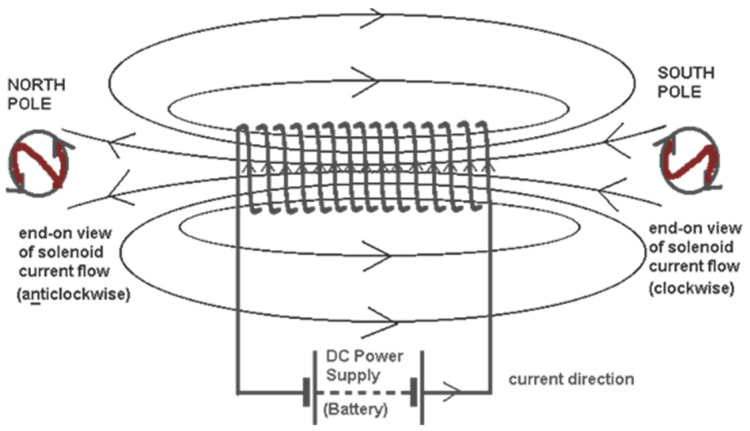
Solenoid Magnetic Field Lines with DC Current [15].

**Figure 2 micromachines-14-00848-f002:**
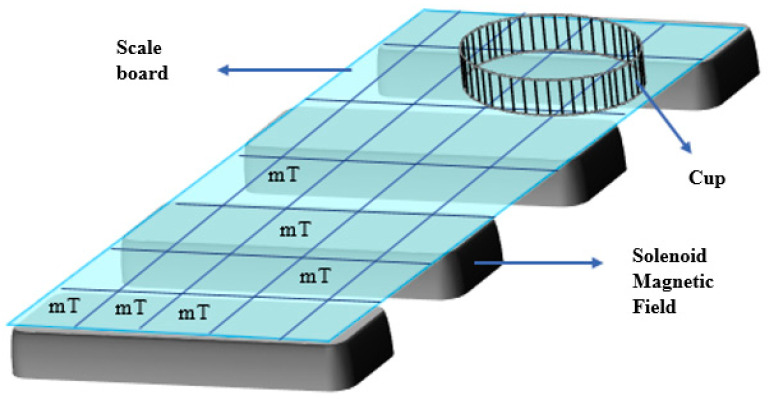
Schematic of scan position characterization of solenoid magnetic field strength.

**Figure 3 micromachines-14-00848-f003:**
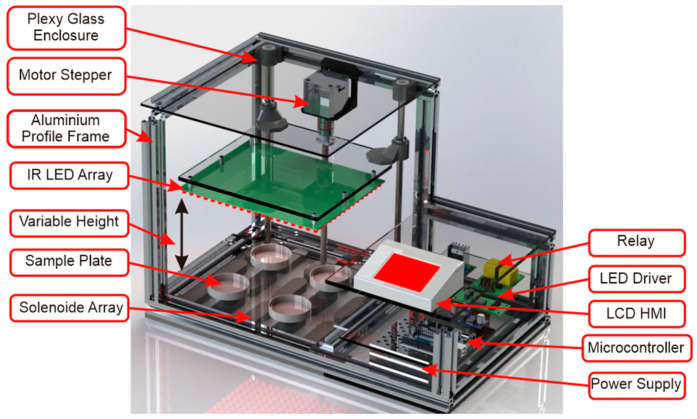
Mechanical Design.

**Figure 4 micromachines-14-00848-f004:**
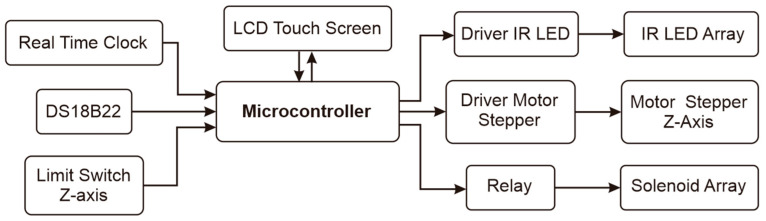
Block diagram system.

**Figure 5 micromachines-14-00848-f005:**
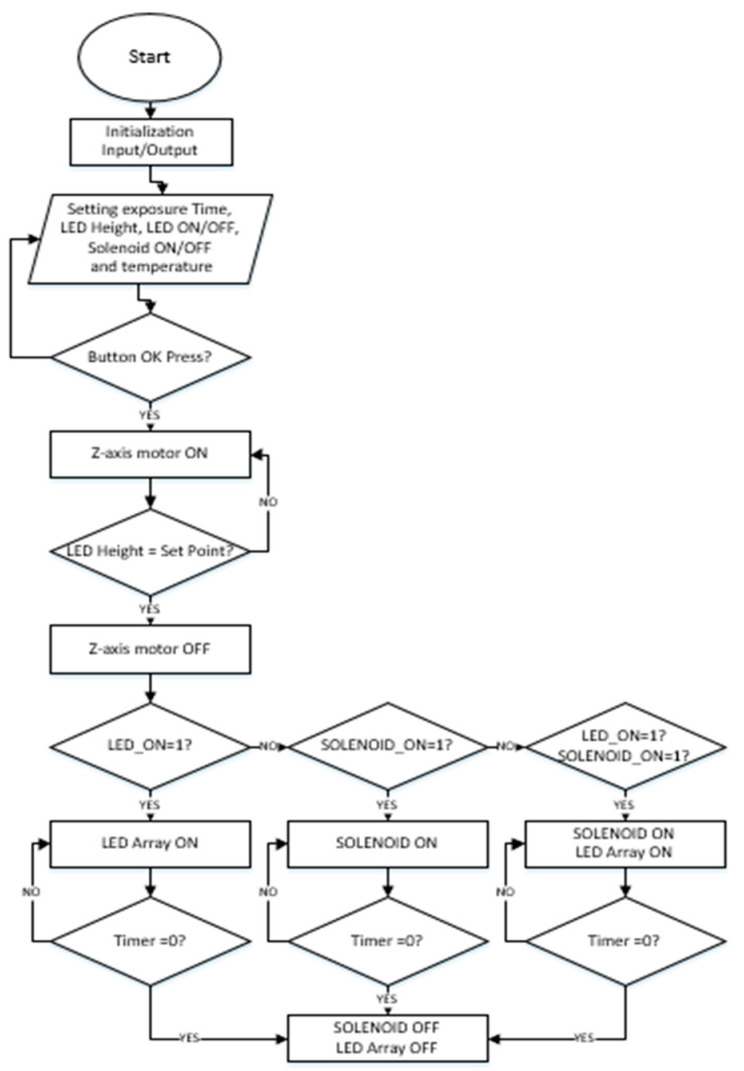
Flowchart software design.

**Figure 6 micromachines-14-00848-f006:**
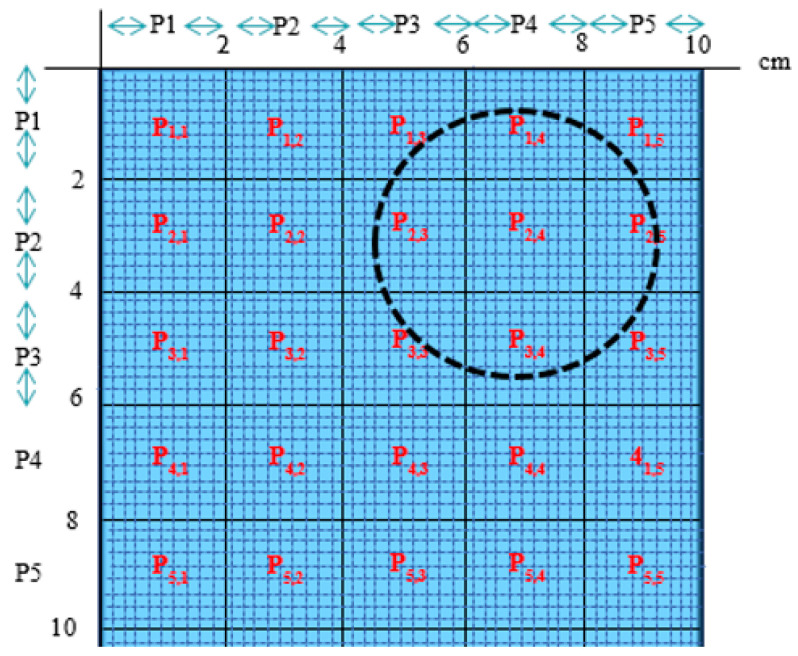
Schematic of the LED power measurement matrix in the sample area.

**Figure 7 micromachines-14-00848-f007:**
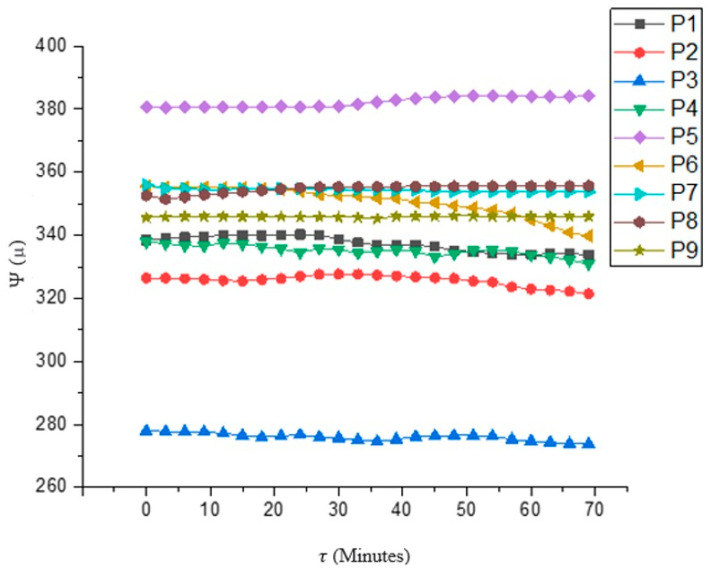
Characterization of infrared LED power with respect to time.

**Figure 8 micromachines-14-00848-f008:**
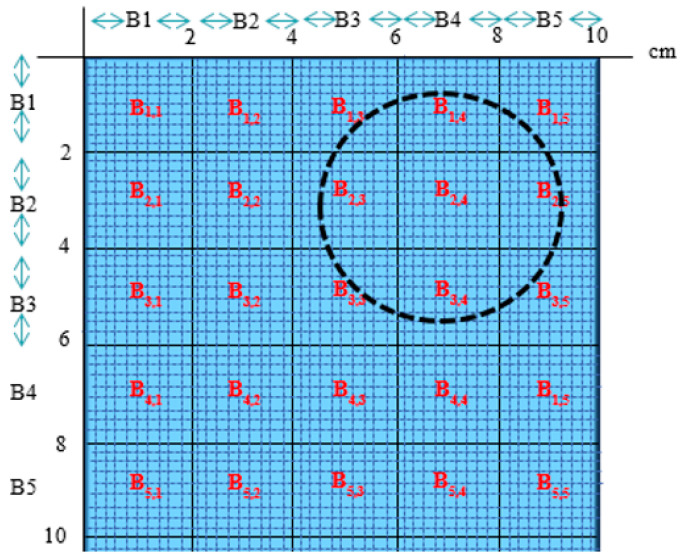
Schematic of the magnetic field strength measurement matrix in the sample area.

**Figure 9 micromachines-14-00848-f009:**
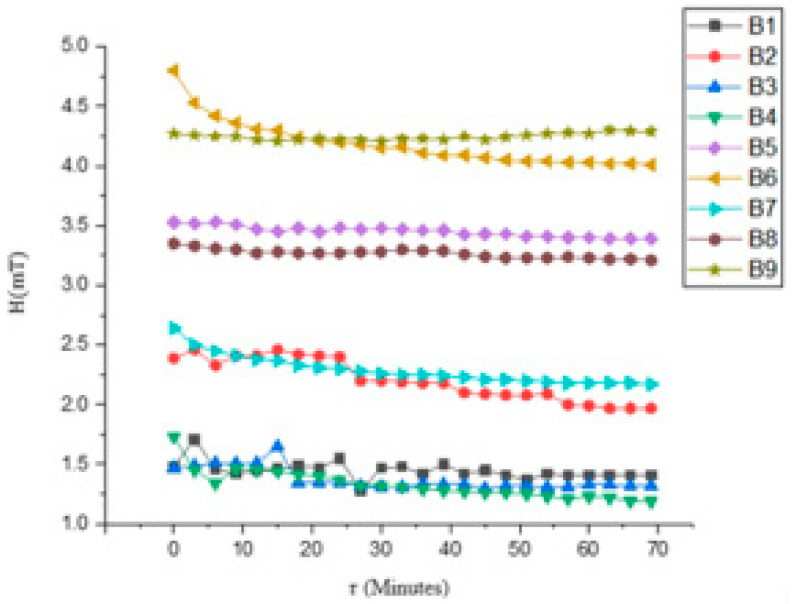
Characterization of the magnetic field of the solenoid with respect to time.

**Figure 10 micromachines-14-00848-f010:**
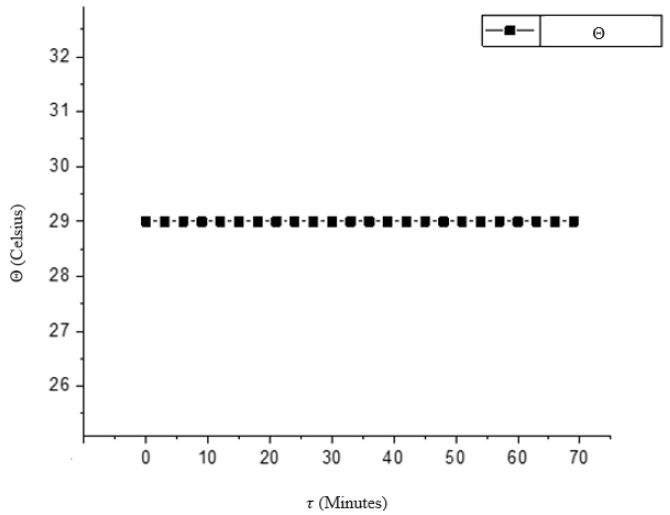
Characterization of irradiation temperature against time.

**Figure 11 micromachines-14-00848-f011:**
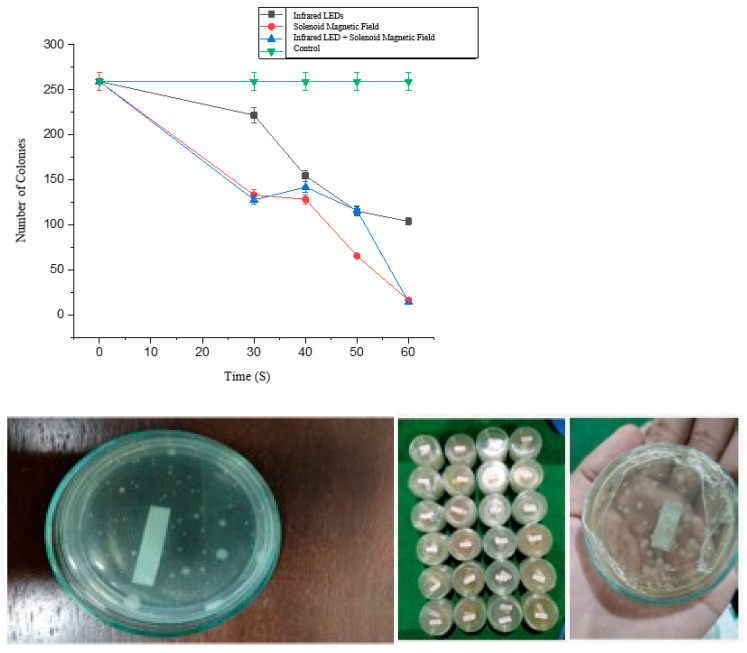
Graph of the decrease in the number of *S. aureus colonies* with time from treatment results with the original agar plate containing the number of colonies.

**Figure 12 micromachines-14-00848-f012:**
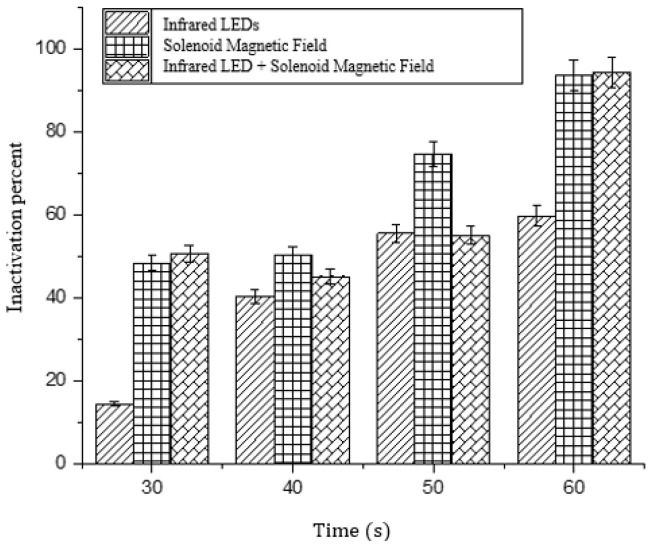
Graph of the percentage of inactivation of *S. aureus* against time from treatment results.

**Figure 13 micromachines-14-00848-f013:**
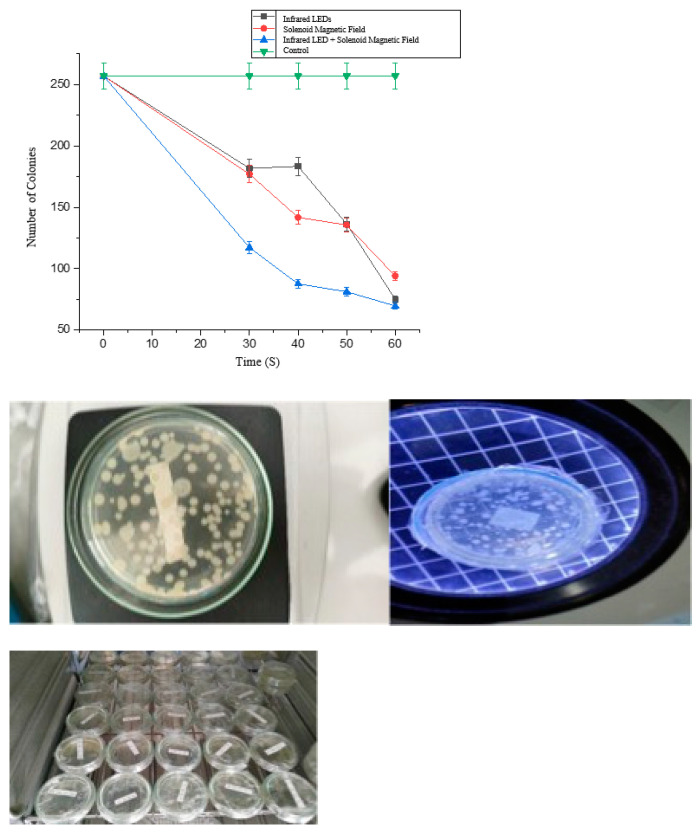
Graph of the decrease in the number of *E. coli* colonies with time from treatment results with the original agar plate containing the number of colonies.

**Figure 14 micromachines-14-00848-f014:**
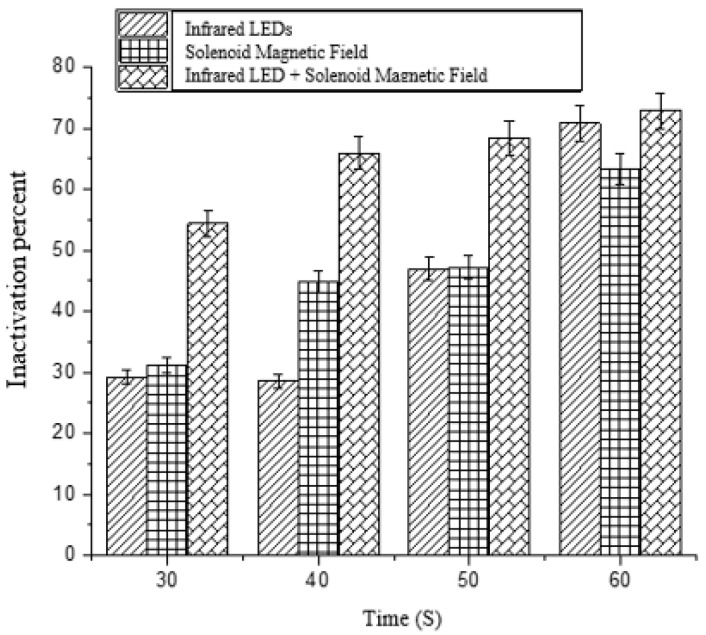
Graph of the percentage of inactivation of bacteria *E. coli* against time from treatment results.

**Table 1 micromachines-14-00848-t001:** Table of rated power in the sample room.

Rated Power (µW)								
P3,1	P3,2	P3,3	P4,1	P4,2	P4,3	P5,1	P5,2	P5,3
384.42	368.96	333.13	381.15	391.56	346.56	355.24	344.73	327.80

**Table 2 micromachines-14-00848-t002:** Intensity and Energy Dose Values of Infrared LEDs.

∑P(W)	$A\left(cm^{2}\right)$	I (Wcm2)	t(Minutes)	D (Jcm2)
**0.0323355**	19.625	0.000164767	30	0.297
			40	0.395
			50	0.494
			60	0.593

**Table 3 micromachines-14-00848-t003:** Table of the distribution of the magnetic field strength of the solenoid in the sample room.

Rated Power (mT)								
B3,1	B3,2	B3,3	B4,1	B4,2	B4,3	B5,1	B5,2	B5,3
0.82	0.16	0.28	3.28	3.09	1.63	5.23	6.13	2.30

**Table 4 micromachines-14-00848-t004:** Table of conclusions of statistical test results on *Staphylococcus aureus*.

Time Exposure	Bacterial Inactivation Percentage (%)		
	Infrared LEDs	Solenoid Magnetic Field	Infrared LEDs and Solenoid Magnetic Field
30 ^1^	20.94 ^a^ ± 5.68	46.92 ^a^ ± 2.33	49.37 ^a^ ± 7.60
40 ^1^	38.82 ^a^ ± 6.48	49.67 ^a^ ± 5.49	44.86 ^a^ ± 7.72
50 ^2^	55.09 ^b^ ± 5.23	62.84 ^b^ ± 2.39	54.36 ^b^ ± 6.31
60 ^3^	59.29 ^c^ ± 6.01	93.45 ^c^ ± 2.29	94.43 ^c^ ± 6.63

**Table 5 micromachines-14-00848-t005:** Table of conclusions on statistical test results on *Escherichia coli* bacteria.

Time Exposure	Bacterial Inactivation Percentage (%)		
	Infrared LEDs	Solenoid Magnetic Field	Infrared LEDs and Solenoid Magnetic Field
30 ^1^	29.27 ^a^ ± 3.93	30.65 ^a^ ± 6.05	53.41 ^a^ ± 6.51
40 ^1^	28.28 ^b^ ± 5.61	43.96 ^b^ ± 9.03	65.41 ^b^ ± 6.36
50 ^2^	46.67 ^c^ ± 5.05	46.70 ^c^ ± 6.09	67.82 ^c^ ± 6.26
60 ^3^	71.31 ^d^ ± 7.04	63.14 ^d^ ± 3.43	72.48 ^d^ ± 5.06

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
