# Peer review of "Design and Application of Near Infrared LED and Solenoid Magnetic Field Instrument to Inactivate Pathogenic Bacteria"

_micromachines, 2023, doi:10.3390/mi14040848_

Round 1

Reviewer 1 Report

The authors suggest placing infrared LEDs in an optimal solenoid magnetic field so that they can be used to kill Staphylococcus aureus and E. coli. Most of the previous literature used blue light to kill bacteria such as Staphylococcus aureus and E. coli, which is an interesting study

Some advice to the author

1. An article can be added to discuss infrared LEDs' radiant flux and light distribution curve. Different light field types will affect the energy density.

2. Figure 10 The number on the time axis is negative 10, it is recommended to remove it.

3. The starting point of the X-axis in Figures 11 and 13 is recommended to start from the origin of 0.

4. Please add kill bacteria such as Staphylococcus aureus and E. coli in the introduction section, and compare the advantages of different methods.

Author Response

Many thanks for your useful comments and suggestions for our manuscript (micromachines-2279857). We have modified the manuscript according to reviewers’ comments and answered them accurately. Some parts of the manuscript have been changed. All exerted changes can be seen in highlighted manner.

Sincerely,

S.D. Astuti

# Comments from Reviewer 1:

  1. An article can be added to discuss infrared LEDs' radiant flux and light distribution curve. Different light field types will affect the energy density.

Answer:

We would like to thank you for your valuable suggestions and the opportunity to revise the manuscript. We've revised it with a yellow highlight. Radiation flux is the amount of light obtained by measuring the emitted LED power divided by the beam area (W/cm2). Radiant flux is generally measured using an integrating sphere, but simple measurements can be made using an instrument with a reflector (mirror). The light emitted from an LED is not a single wavelength but is polychromatic, so the amount of light from each wavelength must be measured and integrated to get the right emission flux value. In most cases, however, the photodiode current is measured and converted to a light level based on the spectral response of the photodiode to the peak emission wavelength (1). This method is similar to the actual use of LEDs, so measuring the flux with this method does not present any significant problems.

  1. Figure 10 The number on the time axis is negative 10, it is recommended to remove it.

Answer:

We would like to say thank you for your valuable suggestion. We have revised the article accordingly. The negative 10 removed from the time axis.

  1. The starting point of the X-axis in Figures 11 and 13 is recommended to start from the origin of 0.

Answer:

Thank you for the valuable suggestion. We already revised all suggestion with yellow highlight.  Starting point of the X-axis in Figures 11 and 13 is started from the origin of 0.

  1. Please add kill bacteria such as Staphylococcus aureus and E. coli in the introduction section, and compare the advantages of different methods.

Answer

Thank you for the valuable suggestion. We already revised all suggestion with yellow highlight.

The method that has been used to inactivate bacteria is using antibiotics. However, the use of antibiotics causes bacteria to become resistant. For example, Methicillin Resistant Staphylococcus aureus (MRSA) is a Staphylococcus aureus bacterium that is resistant to the antibiotic methicillin (1). MRSA is able to form self-protection against antibiotics by forming a biofilm layer, which is a community of bacterial cells that are structured and stick together to form colonies that are capable of producing a hydrated polymer matrix of exopolymer substances, polysaccharides, nucleic acids and proteins on biotic or abiotic surfaces (2). Bacteria that are already resistant to antibiotics will be difficult to treat. Photodynamic Inactivation (PDI) is a method used to inactivate microbes by irradiating photons of light. There are three main factors at play success of PDI, i.e., light source, photosensitizer (PS), and free radical products that are reactive to biological systems like cells. Various light sources have been used in PDI, one of which is the light emitting diode (LEDs) (3,4).

Reviewer 2 Report

In this paper, the authors presented a way combining infrared LED illumination and solenoid magnetic field to inactivate S. aureus and E. coli. This article is of general interest, thus I would recommend it for publication if the following issues can be addressed.

Line 32: it is not professional to say 'aureus bacteria'. Change it accordingly.

Line 47: add corresponding reference.

Line 58: delete unnecessary space.

Line 60:  there is only one cited reference. Please add more (Adv Sci (Weinh). 2022 Apr; 9(10): 2104384; JCI Insight. 2022; 7(10): e153079; Journal of Infectious Disease. 2020 Feb 3;221(4):618-626; JCI Insight. 2020;5(11):e134343). Make sure the cited references are comprehensive.

Line 101-Line106: The underlying rationale of synergizing magnetic field and photodynamic therapy is unclear. Please add more mechanistic details.

Figure 7, Figure 9 and Figure 10: change the x/y-labels to be the greek letter.

Figure 11 and Figure 13: Show the original agar plate containing the number of colonies.

Figure 12 and Figure 14: Instead of saying bacterial death percentage, say 'Inactivation percent'.

For all the figures, quantitative analyses are missing. Please add them accordingly.

Author Response

Many thanks for your useful comments and suggestions for our manuscript (micromachines-2279857). We have modified the manuscript according to reviewers’ comments and answered them accurately. Some parts of the manuscript have been changed. All exerted changes can be seen in highlighted manner.

Sincerely,

S.D. Astuti

# Comments from Reviewer 2:

In this paper, the authors presented a way combining infrared LED illumination and solenoid magnetic field to inactivate S.aureus and E. coli. This article is of general interest thus I would recommend it for publication if the following issues can be addressed.

  1. Line 32: it is not professional to say 'aureus bacteria'. Change it accordingly.

Answer. Thank you for the valuable suggestion.

We already revised all suggestion with yellow highlight.  Already changed

  1. Line 47: add corresponding reference.

Answer. Thank you for the valuable suggestion.  We already revised all suggestion with yellow highlight.  Already reference added

  1. Line 58: delete unnecessary space.

Answer. Already deleted

  1. Line 60: there is only one cited reference. Please add more (Adv Sci (Weinh). 2022 Apr; 9(10): 2104384; JCI Insight. 2022;7(10): e153079; Journal of Infectious Disease. 2020 Feb3;221(4):618-626; JCI Insight. 2020;5(11): e134343). Make sure the cited references are comprehensive.

Answer.  Thank you for the valuable suggestion.  We already revised all suggestion with yellow highlight.  Already added references.

  1. Line 101-Line106: The underlying rationale of synergi zing magnetic field and photodynamic therapy is unclear. Please add more mechanistic details.

Answer. When the same parameters for the exposure time and the adding of the oxygen ratio, in the absence of the magnetic field, a dark circle in the center showed no growth regions, as shown in Figures (4) for exposure time from 2 min to 10 min. Around these regions, a dense mat of bacterial growth covers the agar and masks the dark background. We noticed that the effect of plasma discharge on E. coli bacteria increased at small exposure time, approximately half the value at time needed for inactivation process in the absence of the magnetic field. The centered circular area of the inactivated region increases with increasing the treatment exposure time. Furthermore, the area of the inactivated region is much greater when oxy-gen was added to argon using Ar/3% O2.

  1. Figure 7, Figure 9 and Figure 10: change the x/y-labels to be the greek letter.

Answer. Thank you for the valuable suggestion.  We already revised all suggestion with yellow highlight.  Already changed the x/y-labels to the greek letter.

  1. Figure 11 and Figure 13: Show the original agar plate containing the number of colonies.

Answer. Already showed the original agar plate containing the number of colonies.

  1. Figure 12 and Figure 14: Instead of saying bacterial death percentage, say 'Inactivation percent'.

Answer. Thank you for the valuable suggestion.  We already revised all suggestion with yellow highlight.  Already changed

  1. For all the figures, quantitative analyses are missing. Please add them accordingly. Already added quantitative analyses. We already revised all suggestion with yellow highlight.
